# Emotional Status, Motor Dysfunction, and Cognitive Functioning as Predictors of Quality of Life in Physically Engaged Community-Dwelling Older Adults: A Structural Equation Modeling Approach

**DOI:** 10.3390/ijerph21111469

**Published:** 2024-11-04

**Authors:** Inaihá Laureano Benincá, Maria Gattuso, Stefania Butti, Davide Caccia, Francesca Morganti

**Affiliations:** 1Department of Human and Social Sciences, University of Bergamo, 24129 Bergamo, Italy; maria.gattuso@unibg.it (M.G.); stefania.butti@unibg.it (S.B.); francesca.morganti@unibg.it (F.M.); 2University Sport Centre—CUS, University of Bergamo, 24044 Bergamo, Italy; davide.caccia@mail.cusbergamo.it; 3Centre for Healthy Longevity—CHL, University of Bergamo, 24129 Bergamo, Italy

**Keywords:** group exercise, active aging, mental health, SEM, quality of life

## Abstract

The demographic transition has become a reality, and it demands public policies to promote physical and mental health in aging. Group exercise emerges as a cost-effective and accessible alternative to promote active aging on a large scale, but to optimize the effectiveness of these programs, it is crucial to understand the underlying mechanisms that improve quality of life. This study aimed to explore the associations between emotional status, cognitive functioning, motor dysfunction, and their relationship with quality of life in community-dwelling older adults participating in a group physical exercise program. Structural equation modeling was used to explore these relationships in a sample of 190 older adults. Emotional state directly predicted all domains of quality of life. Motor dysfunction predicts the physical health domain of quality of life. Motor dysfunction and cognitive functioning are strongly correlated with emotional status. The fit indices of the final model are acceptable, and it demonstrates that within group-exercise dynamics, emotional status is the main component of quality-of-life promotion. Therefore, professionals designing group physical exercise programs to promote active aging should consider not only physical fitness, but also the integration of psychosocial elements, offering a holistic approach to enhancing overall well-being.

## 1. Introduction

The demographic transition of recent years has led to an ever-increasing number of older people [1], forcing us to think more about healthy and active aging so that we can manage and adjust policies to support and care for the aging population in the coming years.

Active aging means maximizing opportunities for health, participation, and security to enhance quality of life (QoL) as people age [2]. Furthermore, as people grow older, their QoL is closely related to their levels of independence, autonomy, and healthy life expectancy [3].

Today, neoliberal approaches to healthy longevity are heavily debated, as these place the responsibility for aging at the individual and not the societal level [4]. This is problematic because a major barrier to successful aging is inequality. For example, older adults with lower socioeconomic status tend to have poorer physical health [5] and engage in less physical activity [6,7]. This highlights the importance of public investment in providing accessible programs for healthy aging, such as physical exercise programs for community-dwelling older adults, in order to promote active and healthy aging.

Physical exercise has been demonstrated to delay the progression of chronic diseases and reduce associated risks [8]. Additionally, it enhances the functional capacity and independence of older people, thus preventing falls [9]. Particularly when performed in group settings, exercise has positive effects on mental well-being, as social participation and physical activity can mutually reinforce each other [10]. Community-based group exercise has been proven to increase muscle strength, balance, overall function [11], and health-related QoL [12]. These findings support the current promotion of social policies that include exercise as a way to preserve well-being across the lifespan, especially in a scenario where demographic transition has become a reality [1].

The interaction between the above-mentioned domains in which exercise has positive influences is clearly multidimensional. However, to date, the literature has not clarified which factors of physical health and engagement in physical activity can influence change to the point of improving the daily living conditions of older people.

Furthermore, physical exercise seems to support various reserves in later life. It preserves neuronal structural integrity and brain volume, thus supporting cognitive reserve [13]. Additionally, it serves as a means to build social relationships, helping to promote engagement in activities and consequently social reserve [14]. Preliminary evidence also indicates its role on reserves at the cerebral, cerebellar, and muscular level [15]. These factors serve as proxies for the development of a protective lifestyle that minimizes frailty in aging while maximizing resilience [16].

This multidimensional nature of QoL is also evident in the context of enriching QoL as one ages. Maintaining a high QoL as one ages is considered an important factor in healthy longevity and influences many of the trajectories of aging that can define an older person as active and successful. From a global perspective, QoL represents an individual’s cognitive assessment of their satisfaction with their life in relation to their value system, goals, expectations, standards, and concerns [17,18]. This definition highlights the importance of cognitive evaluation in understanding QoL, as it plays a role in deriving meaning from experiences to build motivation [19], which is essential for empowering the aging population [20], and overall life satisfaction as a consequence [21]. Additionally, physical and emotional evaluation can provide valuable insights, especially when considering health-related QoL, which is just as important as from athe global perspective [22]. Health-related QoL is usually conceptualized in terms of self-perceived health status and physical and emotional function [23].

Studying the role of QoL in healthy longevity is complex and requires a comprehensive analytical approach, such as structural equation modeling. Previous studies have proposed constructing models to predict QoL, with physical fitness being a significant predictor [22,24,25]. However, few studies have specifically investigated physically active older adults with a particular focus on psychological empowerment [19,21].

Therefore, this study aims to enrich the theoretical framework by exploring the associations between emotional status, motor dysfunction, cognitive functioning, and QoL (encompassing the domains of physical health, psychological well-being, social relationships, and environment) in community-dwelling older adults participating in a group physical exercise program. Based on the literature mentioned above, we hypothesize that emotional status, motor dysfunction, and cognitive functioning will be significantly interrelated and will each have an impact on overall QoL.

Building a model that targets active older adults can help to clarify the underlying mechanisms of maintaining QoL through regular group exercise, considering its multidimensional nature. This approach can significantly enhance the effectiveness of disease prevention and active aging promotion programs for healthy longevity.

## 2. Materials and Methods

### 2.1. Study Type

This is a cross-sectional, observational study conducted in collaboration with the University of Bergamo Sport Center (CUS).

### 2.2. Participants

Three hundred and thirty-nine community-dwelling individuals over the age of 60 years from eight municipalities in Bergamo province were eligible to participate in the study. Participants were recruited on a voluntary basis through the Adapted Physical Activity Program (AFA). Individuals using pacemakers were excluded.

This study was conducted in accordance with the Declaration of Helsinki, and it was approved by the Ethics Committee of the University of Bergamo (Approval Number 2024_04_02). All participants were informed (both written and orally) about the study’s objective, methods, and risks. They were also informed about the data collection procedure and gave written informed consent to participate before entering the study. Participant recruitment took place from 13 to 31 May 2024.

### 2.3. Adapted Physical Activity Program

Participants take part in groups of about 20 people and perform 60 min of structured exercise twice a week under the supervision of a licensed sports trainer. Each session comprises 20 min of warm-up exercise, followed by 30 min of light-to-moderate intensity global strengthening and aerobic exercise, as well as dual-tasking and balancing activities, ending with 10 min of cooling-down activities. All participants received medical clearance before taking part in this program.

### 2.4. Outcome Measurements

Demographic data, anthropometric measurements, participants’ medical history, and the duration of participation in the program were recorded. All motor assessments were conducted by a physiotherapist, while the psychological assessments were performed by a psychologist, both with more than five years of experience.

#### 2.4.1. Motor Assessment

The Two-Minute Step Test (2MST) was performed to measure endurance. The number of times the right knee completed full steps in two minutes was counted. A full step was considered when the knee was raised midway between the patella and the iliac crest. Participants were instructed to complete as many steps as possible without straining, and rest was allowed. Participants were notified of the time 1 min and 30 s before test completion [26,27]. The test was performed close to a wall, and heart rate was measured during its performance for security (Polar RS800CX RUN, Polar Electro Oy, Kempele, Finland).

The Timed Get Up and Go (TUG) test was performed to assess mobility and its underlying determinants (strength and balance). The test involves measuring the time participants require to rise from a chair, walk three meters, turn after crossing a line, return, and sit down again; participants were closely monitored during the test’s execution [28]. Upper-limb strength and function were assessed using the 30 s Arm Curl (30AC) and Hand Grip Strength (HGS) tests. In the 30AC test, the number of times a participant curls a hand weight in 30 s through the full range of motion is counted; 2 kg was used for women and 4 kg for men [29]. In the HGS test, the participants were first given an explanation of how to use the dynamometer and were then encouraged to squeeze as hard as possible (KPD-EH 101, Shenzhen Chunhui Electronic Commerce, Shenzhen, China). Participants were positioned in a chair with their wrist in a neutral position and their elbow bent at a 90° angle; three readings with intervals of at least 30 s were recorded for each hand, and the highest reading was used as the final score [30].

Finally, the Short Physical Performance Battery (SPPB) for older adults was performed and scored according to Silva et al. [31]. This battery of tests evaluates lower-limb function and includes balancing tests (Side-by-Side, Semi-Tandem, and Tandem Stand), the 4 m Gait Speed test, and the 5 Times Sit-to-Stand test.

#### 2.4.2. Psychological Assessment

Executive functioning was assessed using five subtests, namely, the similarities test, phonological verbal fluency, Luria’s fist–edge–palm test, conflicting instructions, and the go-no-go test of the Frontal Assessment Battery (FAB15), which has a cut-off of 9.36 [32]. The WAIS digit span test was used to assess short-term memory. In this test, digit sequences ranging from two to nine digits are used, and the score ranges from zero to nine [33]. The 15-item Geriatric Depression Scale (GDS) was used to assess emotional status, and a cut-off score of ≥5 was adopted as an indicator of depressive symptoms [34]. Lastly, the 26-question World Health Organization Quality of Life Brief Version (WHOQOL-BREF) was used to assess quality of life in four domains: physical health, psychological well-being, social relationships, and environment [35].

### 2.5. Statistical Analysis

The software R 4.3.3 (R Foundation for Statistical Computing, Vienna, Austria) was used to analyze all data, and the sample size was selected based on Bagozzi’s study [36]. Before conducting the analysis, the scores of the FAB15 and the WAIS digit span test were adjusted according to Ilardi et al. [32] and Orsini et al. [33], respectively, while the WHOQOL-BREF raw scores were converted to a 4–20 scale for each domain [35]. Physical capacity and psychological well-being were compiled using descriptive statistics.

Distribution normality was tested using the Shapiro–Wilk test for normality. The quantitative variables included in the analyses were the following: overall physical status, comprising all five motor tests and body mass index (BMI, expressed as kg/m^2^); cognitive functioning, comprising the FAB15 and the WAIS digit span test; and affective symptoms measured as scores derived from the GDS. These represented the dependent variables used in the structural model. Sex was treated as a dummy variable.

Finally, a structural equation model (SEM) was applied to investigate the predictors of QoL. The diagonally weighted least squares (DWLS) estimator was used, given its suitability to deal with a mix of ordinal and continuous data [37]. Observed and latent variables were included in the model, and there were no missing data in the model. The indices to evaluate the goodness of the model were the comparative fit index (CFI) and the Tucker–Lewis index (TLI), with values of ≥0.9, the root mean square error of approximation (RMSEA), and the standardized root mean squared residual (SRMR), with acceptable fit values of 0.07 and 0.08 or less, respectively. Standard errors and standardized beta coefficients were also calculated. The a level was set at ˂0.05.

## 3. Results

A total of 190 older adults (31 males) with a mean age of 74.1 years (SD = 6.5) volunteered to participate (Figure 1). Participants’ motor and psychological variables categorized by age group and sex are presented in Table 1 and Table 2, respectively (see Appendix A for the original dataset). The median time of previous participation in the program was 11 (IQR 4–36) months, and median years of education were 8 (IQR 5–13) for females and 13 (IQR 8–13) for males. The most prevalent medical conditions were cardiovascular (43%), musculoskeletal (32%), and metabolic (12%) conditions. Participants’ BMI [38] spanned across underweight (1%), normal weight (36%), overweight (44%), and obesity (18%). Half of the participants were living alone.

Normative values were reached by 25% of participants [39] in the 2MST and by 41% in the TUG test [28]. For the SPPB, data were grouped in three categories: ≤6, between 7 and 9, and >9 [40]. Notably, 10% of participants had a score of <6, 12% had a score between 7 and 9, and 77% had a score of >9. With regard to upper-limb performance, 78% of participants reached normative values in the HGS test [41] and 23% in the 30AC test [27].

With regard to psychological tests, 29% of participants were below the cut-off score of the FAB15, indicating cognitive dysfunction [32], while 13% of participants were above the cut-off point of the GDS, indicating a depressive state [34]. In the evaluation of verbal memory, 84% of participants reached the highest score: between 5.25 and 9. Finally, 50%, 88%, 66%, and 76% of participants reached less than 70% of the maximum score [42] in each of the following domains: physical health, psychological well-being, social relationships, and environment, respectively.

### SEM Model

The hypothesized model, including motor dysfunction, cognitive functioning, and QoL as a latent variable, did not show an appropriate fit (RMSEA 0.067, SRMR 0.075, CFI 0.907, TLI 0.879, and χ^2^
*p* = 0.000). A second model with best fit (Figure 2) was therefore developed, with each domain of QoL considered instead of QoL as a latent variable (RMSEA 0.022, SRMR 0.060, CFI 0.992, TLI 0.987, and χ^2^
*p* = 0.301).

The detailed model results presented in Table 3 highlight that all observed variables were significantly correlated with the latent variables of motor dysfunction and cognitive functioning. Depressive symptoms were negatively correlated with all four domains of QoL, indicating that higher levels of depression are associated with lower levels of perceived QoL across the domains. Motor dysfunction was negatively correlated with the physical health domain, suggesting that higher motor dysfunction implies lower perceived physical health. Cognitive functioning did not influence any domain of perceived QoL.

The residual correlations indicate that cognitive functioning is inversely correlated with depression symptoms (z = −3.491, *p* < 0.001), so better cognition leads to lower depressive symptoms. Motor dysfunction is positively correlated with depression symptoms (z = 3.648, *p* < 0.001) and negatively correlated with cognitive functioning (z = −3.894, *p* < 0.001), indicating that better motor outcomes imply lower depressive symptoms and better cognition.

## 4. Discussion

This study aimed to explore the relationships between four domains of QoL and emotional status, motor dysfunction, and cognitive functioning in community-dwelling older adults participating in group exercise. The proposed model partially supported our hypothesis, indicating that each domain of QoL is directly influenced by emotional status, while motor dysfunction directly influences only the physical domain, and cognitive functioning does not directly influence any domain of QoL directly. However, motor dysfunction and cognitive functioning influence QoL indirectly.

The objective measurements of body function demonstrate that lower-extremity mobility (TUG), lower-extremity function (SPPB), and endurance (2MST) are the measurements with the most impact on the motor dysfunction latent variable (Table 3). This result implies that these aspects of physical functioning are more relevant for identifying overall motor dysfunction in older adults. The same reasoning applies when analyzing the objective measurements of the cognitive functioning latent variable. Executive functioning (FAB15) has a higher impact on cognitive functioning in comparison to short-term memory (WAIS digit spam), as demonstrated in Table 3.

Overall, the participants’ motor evaluations indicated that endurance (2MST), mobility (TUG), and arm strength (30AC) are the most impaired functions, with 75%, 59%, and 77% of non-normative scores, respectively. However, when analyzing HGS, 78% of participants achieved normative values. This is an interesting finding, as hand grip strength has been associated with sarcopenia, as well as disability, morbidity, and mortality rates in older adults [43]. In addition, 77% of participants achieved a score over 9 in the SPPB; scores below this threshold indicate increased odds of institutionalization and increased functional decline or disability [44].

Although most participants did not achieve the normative values for endurance, mobility, and arm strength, it is interesting to note that the majority of participants did not have depressive symptoms or cognitive dysfunction. This finding can be explained by Miller et al. [45], who stated that group physical exercise builds self-efficacy and social support, with the latter being the strongest predictor of depressive symptoms in older adults, independent of caloric expenditure.

Regarding cognitive functioning results, previous studies have demonstrated that the protective effects of exercise are enhanced by participation in groups [46,47], independent of the amount of physical activity undertaken. This can be attributed to the creation of social relationships and improved adherence to exercise [45,48].

Even though the benefits of group exercise on emotional status and cognitive functioning seem to be independent of the amount of physical activity undertaken, it is interesting to note that motor dysfunction was inversely related to the physical domain of QoL. This domain includes aspects such as physical pain, medical treatment, energy levels, mobility, sleep quality, activities of daily living, and capacity for work [18]—aspects that, if hindered, can probably impact both mood and cognition. A previous structural equation model demonstrated that active older adults experienced better sleep quality and overall well-being (encompassing mental, physical, and social aspects), which in turn led to more engagement in physical activity and enhanced health-related QoL in the long term [49].

The relationship between QoL and cognitive functioning is less straightforward. Consistent with our results, previous studies have reported that mood is more closely related to QoL than cognitive functioning [50,51]. Although Munawar et al. [52] found a correlation between cognitive functioning and the WHOQOL-BREF’s domains, the model analyzing cognitive functioning and dependency did not significantly predict QoL. However, a large longitudinal investigation concluded that health-related QoL is negatively associated with cognitive impairment and provided further insights on this discussion [53].

Our study contributes to the existing literature [19,21,22,24,25] by demonstrating that while emotional status directly influences QoL, cognitive functioning and physical activity appear to impact it indirectly through their association with emotional status. When analyzing the residual correlations between motor dysfunction, cognitive functioning, and depressive symptoms, we found that cognitive functioning had the strongest association with depressive symptoms, despite not being directly related to QoL. Additionally, motor dysfunction significantly influenced both cognitive functioning and depressive symptoms, elucidating the dynamic interplay between these factors underlying the direct impact of depressive symptoms on QoL. The observed role of depressive symptoms in QoL aligns with previous findings [54].

This study has some limitations. First, the sample size was a limitation, and future studies should consider replicating this model with more subjects, which would allow for more complex analyses, accounting for the effects of age and sex. While reflecting the majority of women participating in the group exercise program, only 16% of our sample consisted of men, limiting our ability to explore potential associations involving sex and affecting the generalizability of this study. As this is a cross-sectional study, it would be interesting to conduct a longitudinal study to investigate causality within the proposed model and investigate how these variables evolve over time. Additionally, our assessment of emotional status was limited to depressive symptoms only. Even though depressive symptoms were directly correlated with QoL, including other psychological factors, such as anxiety, social isolation, abuse, or fear of falling, would enrich this model. It is also important to highlight that QoL data are not an objective measurement and rely on self-reporting, making it susceptible to recall bias, overreporting, or underreporting. It may also be influenced by varying cognitive levels. Lastly, although socioeconomic and environmental variables were not assessed, the adapted physical activity program is a public initiative for community-dwelling older adults with a symbolic fee, reflecting its homogeneity.

## 5. Conclusions

Our study adds to the literature by showing that when older adults are physically active, emotional status influences all four domains of QoL (physical health, psychological well-being, social relationships, and environment). Meanwhile, physical fitness, measured here as physical dysfunction, only affected the physical health domain of QoL. However, we observed that physical dysfunction and cognitive functioning had a significant relationship with emotional status, thereby indirectly influencing QoL.

In light of the demographic transition, policy strategies have increasingly focused on promoting active aging. One approach to this is the establishment of community-based group exercise. While previous studies have created models to predict older adults’ QoL, often identifying physical fitness as a significant predictor, few have examined this in physically active older adults. The model proposed in this study contributes to a multidisciplinary understanding of the factors promoting QoL by elucidating the interactions between emotional status, motor dysfunction, and cognitive functioning in older adults engaged in community-based group exercise.

Based on the aforementioned results, we suggest that professionals designing group exercise programs as a means to promote active aging should consider, in addition to the acknowledged importance of physical fitness, the integration of psychosocial elements, as these had a major impact on overall well-being. In conclusion, the existing networks established by policymakers to engage older adults in physical exercise present a window of opportunity to further enhance overall well-being and promote healthy aging.

## Figures and Tables

**Figure 1 ijerph-21-01469-f001:**
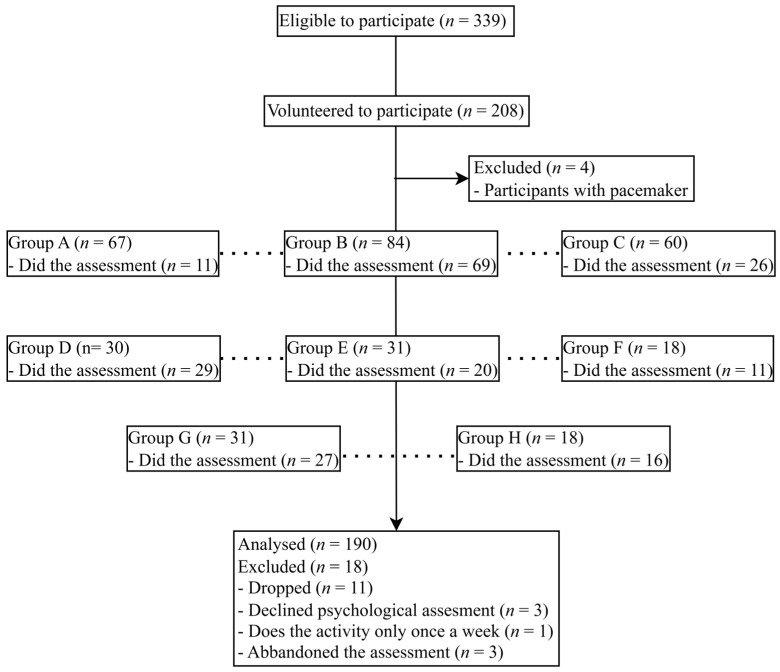
Flow of participants from the eight included municipalities (Groups A–H, connected by dotted lines).

**Figure 2 ijerph-21-01469-f002:**
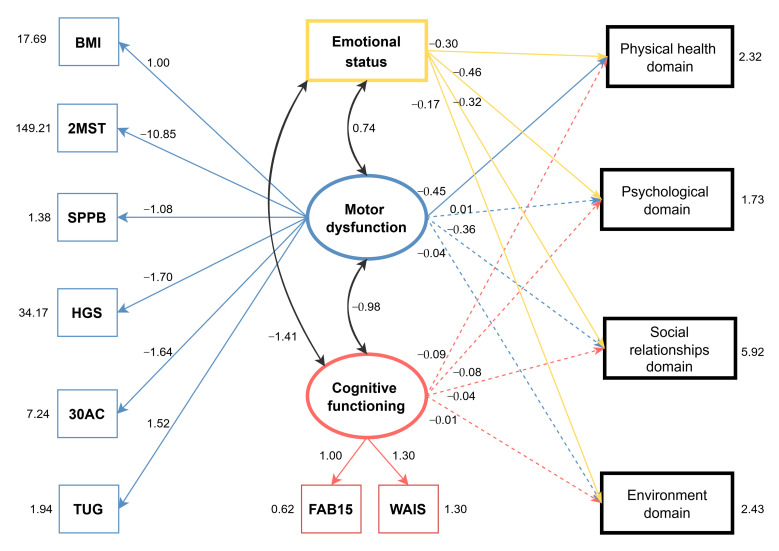
Final model predicting the domains of quality of life. Relations are indicated as unstandardized beta coefficients. Two-sided arrows represent covariances and dashed lines represent non-significant relationships (*p* > 0.05). BMI: body mass index; 2MST: Two-Minute Step; TUG: Timed Get Up and Go; SPPB: Short Physical Performance Battery; HGS: Hand Grip Strength; 30AC: 30 s Arm Curl; WAIS: digit span; FAB15: Frontal Assessment Battery.

**Table 1 ijerph-21-01469-t001:** Participants’ baseline motor characteristics in mean ± SD; median (IQR).

Age Groups	BMI	2MST	TUG	SPPB	HGS	30AC
<70 *n* = 46	26.4 (5.2)	81.8 ± 16.3	10.2 ± 1.6	12 (1.7)	25.5 ± 4	12.9 ± 2.75
Female *n* = 45	26.3 ± 3.9	82.9 ± 14.8	10 ± 1.4	12 (1)	25.5 ± 4	13 ± 2.7
Male *n* = 1	47.5	35	16	9	27	10
70–79 *n* = 101	25.8 (6)	77.4 ± 17.2	10.5 (2.2)	11 (2)	23.5 (6.7)	15.5 ± 3.4
Female *n* = 86	27.1 ± 4.6	75.5 ± 16.6	10.6 (2)	11 (2)	23 ± 4.2	12.3 ± 3.2
Male *n* = 15	26.1 ± 3.9	87.9 ± 17.4	9.6 (2)	12 (0.5)	36.7 ± 6.4	13.2 ± 4.4
>80 *n* = 43	26 ± 2.9	69.7 ± 15.7	12.1 ± 2.1	10 (2)	23.3 (6.6)	11 ± 2.9
Female *n* = 28	25.6 ± 3.2	70 ± 18 ^ǂ^	12.1 ± 2.2	10 (3)	21 ± 3.5	11.4 ± 3
Male *n* = 15	26.9 ± 2.1	69.2 ± 10.8	12 ± 2.2	9.8 ± 1.6	28 (4.3)	10.2 ± 2.7

BMI: body mass index; 2MST: Two-Minute Step; TUG: Timed Get Up and Go; SPPB: Short Physical Performance Battery; HGS: Hand Grip Strength; 30AC: 30 s Arm Curl; ǂ these data exclude a participant that was unable to perform the 2MST.

**Table 2 ijerph-21-01469-t002:** Participants’ baseline psychological characteristics in mean ± SD; median (IQR).

Age Groups	WAIS	FAB15	GDS	Domain 1 ǂ	Domain 2 ǂ	Domain 3 ǂ	Domain 4 ǂ
<70 *n* = 46	6.3 ± 1.1	12 ± 2	2 (3)	15 (2.7)	13.5 (2.7)	15 (3)	14 (1.7)
Female *n* = 45	6.3 ± 1.1	12.1 ± 2	2 (3)	15 (2)	13 (3)	15 (3)	14 (1)
Male *n* = 1	5.7	9.8	1	12	14	13	13
70–79 *n* = 101	6.4 ± 1.3	12.1 (3.1)	3 (3)	16 (2)	14 (2)	15 (5)	14 (2)
Female *n* = 86	6.3 ± 1.3	11.6 ± 2.3	3 (3)	15 (2.7)	14 (2)	14 (5)	14 (2)
Male *n* = 15	6.7 ± 1.2	12.5 ± 1.7	1 (3)	17 (3)	14.9 ± 1.5	16 (1)	15.5 ± 1.6
>80 *n* = 43	6.4 ± 1.1	10.9 ± 2.7	3.6 ± 2.4	15 (2)	14 (2)	12 (4)	15 (2.5)
Female *n* = 28	6.4 ± 1.1	11.1 ± 2.7	3.7 ± 2.2	15 ± 1.4	13.7 ± 1.2	12 (4)	14.5 ± 1.9
Male *n* = 15	6.3 ± 1.1	10.5 ± 2.8	3.4 ± 2.8	15.3 ± 1.4	14.7 ± 2.1	12.1 ± 3.3	15.3 ± 1.4

WAIS: digit span; FAB15: Frontal Assessment Battery; GDS: Geriatric Depression Scale; ǂ these data are from WHOQOL-BREF, in which Domain 1 represents physical health; Domain 2 represents psychological health; Domain 3 represents social relationships; and Domain 4 represents environment.

**Table 3 ijerph-21-01469-t003:** Measurements of the model.

**Structural Model**	**Raw β**	**Standard Error**	**z-Value**	***p*-Value**	**Standardized β**
Physical health					
Depressive symptoms	−0.296	0.091	−3.260	0.001 *	−0.396
Motor dysfunction	−0.448	0.167	−2.689	0.007 *	−0.279
Cognitive functioning	−0.089	0.087	−1.022	0.307	−0.113
Psychological					
Depressive symptoms	−0.460	0.101	−4.562	<0.001 *	−0.460
Motor dysfunction	0.014	0.145	0.099	0.921	0.009
Cognitive functioning	−0.083	0.092	−0.907	0.364	−0.111
Social relationships					
Depressive symptoms	−0.321	0.104	−3.098	0.002*	−0.291
Motor dysfunction	−0.357	0.201	−1.777	0.076	−0.150
Cognitive functioning	0.038	0.118	0.322	0.747	0.033
Environment					
Depressive symptoms	−0.166	0.104	−0.423	0.672	−0.030
Motor dysfunction	−0.044	0.104	−0.423	0.672	−0.030
Cognitive functioning	−0.004	0.067	−0.053	0.958	−0.005
**Measurement Model**	**Raw β**	**Standard Error**	**z-Value**	** *p* ** **-Value**	**Standardized β**
Motor dysfunction					
Body mass index	1.000				0.254
Two-Minute Step Test	−10.854	2.463	−4.407	<0.001 *	−0.700
Short Physical Performance Battery	−1.082	0.253	−4.281	<0.001 *	−0.712
Hand Grip Strength test	−1.697	0.445	−3.813	<0.001 *	−0.305
30 s Arm Curl	−1.644	0.390	−4.214	<0.001 *	−0.559
Timed Get Up and Go	1.523	0.351	4.346	<0.001 *	0.770
Cognitive functioning					
Frontal Assessment Battery 15	1.000				0.943
WAIS digit span	0.187	0.068	2.759	0.006 *	0.346

β: beta coefficients; * *p*-values < 0.05.

## Data Availability

The original contributions presented in the study are included in the article/Appendix A; further inquiries can be directed to the corresponding author/s.

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
