# Peer review of "Emotional Status, Motor Dysfunction, and Cognitive Functioning as Predictors of Quality of Life in Physically Engaged Community-Dwelling Older Adults: A Structural Equation Modeling Approach"

_ijerph, 2024, doi:10.3390/ijerph21111469_

Round 1
Reviewer 1 Report
Comments and Suggestions for Authors
This manuscript explores the associations among emotional status, motor dysfunction, cognitive functioning, and quality of life (QoL) in community-dwelling older adults participating in a group physical exercise program, using a Structural Equation Modeling (SEM) approach. The topic is both interesting and significant. However, I have the following suggestions:
1. In Tables 1 and 2, participants' baseline motor and psychological characteristics are stratified by age and sex. However, in the SEM model presented in Figure 2, age and sex do not appear to have been controlled for. Given their known importance in relation to both predictors and outcomes, I recommend including these variables in the SEM model.
2. In the discussion section (Lines 292-293), the authors state: “Our study contributes to the existing literature by demonstrating that emotional status mediates the relationship between cognitive impairment, physical activity, and QoL.” This statement is inaccurate. Mediation analysis should be performed if the authors intend to draw conclusions about the mediating effect of emotional status. Based on the current analyses, mediation effects cannot be established.
Author Response
Comments 1: In Tables 1 and 2, participants' baseline motor and psychological characteristics are stratified by age and sex. However, in the SEM model presented in Figure 2, age and sex do not appear to have been controlled for. Given their known importance in relation to both predictors and outcomes, I recommend including these variables in the SEM model.
Response 1: We agree with this comment and have attempted to include age and sex in the current model. As age and sex are used to adjust the scores of the FAB15 and WAIS Digit Span, we incorporated these variables into the creation of the latent variables "motor dysfunction" and "emotional status". However, given the increased complexity of the revised model, the software reported problems with model convergence. We believe that the higher number of parameters relative to the sample size made it difficult for the algorithm to find an optimal solution, leading to unreliable parameter estimates. Therefore, we include this as a suggestion for future studies and recognized the sample size as a limitation: “First, the sample size was a limitation, and future studies should consider replicating this model with more subjects, which would allow for more complex analyses, accounting for the effects of age and sex.” - Please check page 8, lines 302-304.
Comments 2: In the discussion section (Lines 292-293), the authors state: “Our study contributes to the existing literature by demonstrating that emotional status mediates the relationship between cognitive impairment, physical activity, and QoL.” This statement is inaccurate. Mediation analysis should be performed if the authors intend to draw conclusions about the mediating effect of emotional status. Based on the current analyses, mediation effects cannot be established.
Response 2: Thank you for pointing this out. We agree that this statement goes beyond what the analysis can support. Therefore, we have rewritten and now it reads: “Our study contributes to the existing literature [19,21,22,24,25] by demonstrating that while emotional status directly influences QoL, cognitive functioning and physical activity appears to do it indirectly, through their association with emotional status” - Please check page 8, lines 292-294.
Reviewer 2 Report
Comments and Suggestions for Authors
Thank authors for identifying evidence for promoting physical activity in older adults living in the community in relation to their domains of quality of life.
To support the quality of the research study, the following is recommended:
I recommend adding the small number of respondents in the research sample to the limitations of the studies.
I recommend adding to the above information in the discussion "Our study contributes to the existing literature by demonstrating that emotional status mediates the relationship between cognitive impairment, physical activity, and QoL" specific citations of research studies that the authors considered to be "the existing literature".
Author Response
Comments 1: I recommend adding the small number of respondents in the research sample to the limitations of the studies.
Response 1: We agree with this recommendation. As mentioned in the text, the sample size was based on the study of Bagozzi [36], who states that: “one should endeavor to achieve a sample size above 100, preferably above 200.”; therefore, we accept that would be preferable to include more subjects and acknowledged sample size as a limitation - Please check page 8, lines 302-303.
Comments 2: I recommend adding to the above information in the discussion "Our study contributes to the existing literature by demonstrating that emotional status mediates the relationship between cognitive impairment, physical activity, and QoL" specific citations of research studies that the authors considered to be "the existing literature".
Response 2: Thank you for this recommendation, we added the references as recommended. - Please check page 8, line 292.
Reviewer 3 Report
Comments and Suggestions for Authors
Please find attached the comments.
regards

Author Response
Comments 1: Gender Imbalance: The sample is predominantly female (>84%), which limits the generalizability of findings across genders. The study lacks exploration into how gender might influence the relationships between motor dysfunction, emotional status, cognitive function, and quality of life. Discussion can also include this aspect in detail. This can be cited as an important limitation.
Response 1: Thank you for pointing this out. Considering your comment about the lack of exploration into how gender might influence the relationships found, we tried to run a new model including sex and observed a worsening of its fits: RMSEA 0.054, SRMR 0.084, CFI 0.947, TLI 0.917, probably by the imbalance between numbers. Therefore, we have included this aspect as an important limitation, now it reads: “While reflecting the majority of women participating in the group exercise program, only 16% of our sample consisted of males, limiting our ability to explore potential associations involving sex and affecting the generalizability of this study.” Please check page 8, lines 304-307.
Comments 2: Limitation regarding using a Cross-sectional Design: Since this is a cross-sectional study, it cannot establish causality between emotional status, motor function, cognitive function, and quality of life. A longitudinal design could provide insights into causal relationships and how these variables evolve over time.
Response 2: We agree with this comment and would like to clarify that studying how these variables evolve over time is planned for a future study. Moreover, this limitation/recommendation has been revised in the article and now it reads: “As this is a cross-sectional study, it would be interesting to conduct a longitudinal study to investigate causality within the proposed model and investigate how these variables evolve over time.” - Please check pages 8-9, lines 307-309.
Comments 3: Focus on Emotional Status appears restricted: The study only assesses depressive symptoms as a measure of emotional status, which restricts understanding of other psychological factors (e.g., anxiety, loneliness, fear of falling, social isolation, emotional/physical abuse etc.) that could significantly impact quality of life in older adults.
Response 3: We agree with this comment and would like to highlight that this limitation has been revised and now it reads: “Additionally, our assessment of emotional status was limited to depressive symptoms only. Even though depressive symptoms were directly correlated with QoL, including other psychological factors, such as anxiety, social isolation, abuse or fear of falling, would enrich this model.” - Please check page 9, lines 309-312.
Comments 4: Self-Reported Quality of Life: Quality of life is measured via self-reports, which are susceptible to recall bias, over- or under-reporting, and may be influenced by cognitive limitations in older adults and is often reported in many studies. Using additional objective measures or caregiver reports could definitely improve data reliability.
Response 4: We agree with this comment and will consider using more objective measures in future studies. Furthermore, we would like to highlight that this limitation has been rewritten considering your comment, and now it reads “It is also important to highlight that QoL data is not an objective measurement and relies on self-reporting, making it susceptible to recall bias, overreporting, or underreporting, and it may be influenced by varying cognitive levels.” - Please check page 9, lines 312-315.
Comments 5: Absence of Socioeconomic and Environmental Variables: Socioeconomic status, living conditions, and social support structures are not considered, though these factors can impact both quality of life and engagement in physical activities. Including these variables would provide a more comprehensive model.
Response 5: Thank you for this observation. We agree that these variables can indeed impact QoL and engagement in physical activity. Unfortunately, socioeconomic status and social support were not assessed in this study. However, the adapted physical activity program is a public initiative with a symbolic fee, reflecting its homogeneity. Regarding living conditions, our study was limited to older adults living in the community, and the only additional information we gathered was whether participants lived alone, limiting our ability to comprehensively analyze the impact of living conditions, as this is a complex subject. Furthermore, it is not possible to analyze the impact of these factors on physical activity engagement, as our inclusion criteria specified community-dwelling older adults who were already engaged in physical activity. We also acknowledge this aspect as a limitation; please refer to page 9, lines 315–318.
Comments 6: Unexplored Impact of Health Conditions: The study mentions common health conditions (e.g., cardiovascular, musculoskeletal issues) but does not examine how these might interact with motor or cognitive function, emotional status, and quality of life, which could be valuable for targeted interventions.
Response 6: Thank you for pointing this out. This is indeed an interesting aspect to consider; however, analyzing the effects of health conditions was not part of our initial hypothesis. We did not control for it as an inclusion criterion, as our objective was to include 'healthy' older adults rather than those needing rehabilitation care (all participants received medical clearance before joining the program). Additionally, the health conditions reported in the descriptive analysis were self-reported, which would not validate their inclusion in the SEM.